# DFT-Spread Spectrally Overlapped Hybrid OFDM–Digital Filter Multiple Access IMDD PONs

**DOI:** 10.3390/s21175903

**Published:** 2021-09-02

**Authors:** Abdulai Sankoh, Wei Jin, Zhuqiang Zhong, Jiaxiang He, Yanhua Hong, Roger Giddings, Jianming Tang

**Affiliations:** School of Computer Science and Electronic Engineering, Bangor University, Bangor LL57 1UT, UK; eep81a@bangor.ac.uk (A.S.); w.jin@bangor.ac.uk (W.J.); eeu970@bangor.ac.uk (J.H.); y.hong@bangor.ac.uk (Y.H.); r.p.giddings@bangor.ac.uk (R.G.); j.tang@bangor.ac.uk (J.T.)

**Keywords:** orthogonal frequency division multiplexing (OFDM), digital filter multiple access (DFMA), DFT-spread OFDM, intensity modulation and direct detection (IMDD), passive optical network (PON)

## Abstract

A novel transmission technique—namely, a DFT-spread spectrally overlapped hybrid OFDM–digital filter multiple access (DFMA) PON based on intensity modulation and direct detection (IMDD)—is here proposed by employing the discrete Fourier transform (DFT)-spread technique in each optical network unit (ONU) and the optical line terminal (OLT). Detailed numerical simulations are carried out to identify optimal ONU transceiver parameters and explore their maximum achievable upstream transmission performances on the IMDD PON systems. The results show that the DFT-spread technique in the proposed PON is effective in enhancing the upstream transmission performance to its maximum potential, whilst still maintaining all of the salient features associated with previously reported PONs. Compared with previously reported PONs excluding DFT-spread, a significant peak-to-average power ratio (PAPR) reduction of over 2 dB is achieved, leading to a 1 dB reduction in the optimal signal clipping ratio (CR). As a direct consequence of the PAPR reduction, the proposed PON has excellent tolerance to reduced digital-to-analogue converter/analogue-to-digital converter (DAC/ADC) bit resolution, and can therefore ensure the utilization of a minimum DAC/ADC resolution of only 6 bits at the forward error correction (FEC) limit (1 × 10^−3^). In addition, the proposed PON can improve the upstream power budget by >1.4 dB and increase the aggregate upstream signal transmission rate by up to 10% without degrading nonlinearity tolerances.

## 1. Introduction

To effectively cope with the current avalanche of mobile traffic, driven by the unprecedented increase in users’ demands for ultrawide bandwidth multimedia and cloud services that have become ubiquitous, fronthauls/backhauls of 5G CRANs capable of converging optical and wireless networks are needed, and require significant changes to network access in order to support the ambitious system requirements [1,2]. Addressing such technical challenges requires multipronged efforts in different network domains across all layers to not only accommodate the explosive expansion in traffic demand, but also efficiently support dynamic bandwidth provisioning, improved cost-effectiveness, and power efficiency [3]. The emergence of software-defined networks (SDNs) with their extended network-control functionalities leverages the abstraction of different physical resources, and enhances the dynamic reconfigurability, flexibility, scalability, and elasticity of the network [4]. To deliver an SDN-based solution capable of satisfying the abovementioned requirements, a non-incremental solution should be implemented to realize the highly desirable, cost-effective, separately implemented, and independently operated legacy optical and wireless access networks in a converged manner. For cost-sensitive application scenarios such as optical access networks, metropolitan area networks (MANs), and mobile fronthaul/backhaul networks, intensity modulation and direct detection-based passive optical networks (IMDD PONs) are considered to be a competitive technical solution due to their excellent cost-effectiveness and power efficiency [5].

To overcome the abovementioned technical challenges, a novel PON technique termed hybrid OFDM–DFMA PON, utilizing spectrally overlapped digital orthogonal filtering, has recently been proposed and extensively investigated [6] where, regardless of the ONU count, matching filter (MF)-free single fast Fourier transform (FFT) operation and the relevant DSP processes are applied in a pipeline approach. In the proposed PON, transceiver-embedded software-reconfigurable digital orthogonal filtering is utilized in each individual ONU, where for upstream transmission two spectrally overlapped digitally filtered orthogonal ONU OFDM sub-band signals occupy the same subwavelength spectral region. In the OLT, for ONU sub-band signal demultiplexing and recovery, a procedure similar to the previously reported hybrid OFDM–DFMA PONs [7] is adopted. Numerical results show that in terms of improving the upstream signal transmission capacity and enhancing the spectral efficiency, the proposed PON outperforms the results previously reported in [7] by a factor of ~2. Moreover, in the context of low OLT-DSP complexity, robustness against practical transceiver impairments, enhanced flexibility, and backward compatibility with existing 4G networks, the proposed PON still maintained all the aforementioned salient unique features associated with the hybrid OFDM–DFMA PONs [6].

It is important to mention that the PONs reported in [6] are OFDM-based. The OFDM’s multi-subcarrier modulation scheme is well known to produce high PAPRs due to the coherent superposition of orthogonal subcarriers in the time domain [8]. Technically speaking, systems with large PAPRs not only require wide dynamic operating ranges for the transceiver-embedded electrical/optical devices, but also produce high quantization noise for a fixed number of quantization bits, and may force the involved devices to operate in their nonlinear regions, thus introducing nonlinear signal distortion. In addition, a large PAPR may also cause serious nonlinear noise associated with standard single-mode fibre (SSMF) nonlinearities [9]. Therefore, suppressing PAPRs in the PONs reported in [6] is of great importance. Several PAPR reduction techniques have been proposed [10,11,12,13]. The most widely adopted and simplest approach is to clip OFDM signals [10,11]; this approach achieves certain levels of PAPR reduction; however, the clipping still causes significant signal distortion. Another solution is to use multiple signalling and probabilistic techniques, such as pilot-assisted partial transmit sequences (PTSs) and selected mapping (SLM) [12,13]. However, due to these approaches requiring redundant information to be transported alongside actual data, they have an intrinsic drawback of reducing the useful data rate and increasing the computational complexity. In contrast to the abovementioned PAPR reduction techniques, DFT-spread OFDM is the ultimate technical solution, because it is free from parallel redundant information and has low complexity, since only deterministic DFT and IDFT operations are required in the transceiver [14]. The DFT-spread OFDM technique has already been reported in SSMF IMDD links, including DFT-spread layered/enhanced asymmetrically clipped OFDM systems [15] and probabilistically shaped OFDM-enabled IMDD systems [16]. In addition, the DFT-spread technique also possesses high compatibility for both long-distance [17] and short-reach IMDD transmission systems [18], and shows superior performance in PAPR reduction. Recently, we have applied the DFT-spread technique in the hybrid OFDM–DFMA PONs to further improve the flexibility of the system transmission performance [19]. However, in these PONs, each individual subwavelength spectral region only conveys either a single in-phase (I) or quadrature-phase (Q) channel upstream of the double sideband (DSB) OFDM signal, which halves the spectral efficiency compared with the spectrally overlapped hybrid OFDM–DFMA PONs [6].

By combining the benefits of the DFT-spread technique and the previously reported PON [6], in this paper, we propose DFT-spread spectrally overlapped hybrid OFDM–DFMA IMDD PONs and, via numerical simulations, analyse and optimize their performance characteristics. The simulation results show that when the DFT-spread technique is applied, PAPR reductions of more than 2 dB are attained for a digitally filtered OFDM signal carrying QAM modulated data. More importantly, the proposed PON can achieve a reduction of more than 1 bit in the minimum required DAC/ADC bit resolution, and an improvement of more than 1.4 dB in the upstream power budget. Furthermore, in comparison to the conventional hybrid OFDM–DFMA PON [6], the proposed PON can enhance the aggregate upstream signal transmission rate by factor of up to 10% in a 25 km SSMF IMDD PON transmission system. It is noteworthy that, while the proposed PON still maintains all of the unique advantages associated with the previously reported PONs [6], in the OLT, without utilizing digital MFs, the single FFT operation followed by summing and subtraction operations of the lower sideband (LSB) and upper sideband (USB), and the corresponding DSP-enabled data recovery processes applied in a pipelined approach, can directly demultiplex and demodulate ONU sub-band signals within the same subwavelength spectral region, while the same OLT receiver can also be used to demodulate legacy OFDM signals.

The above salient features make the proposed PON a feasible solution for future 5G networks in terms of providing DSP-enabled multichannel aggregation and deaggregation solutions for fronthaul networks [20] to effectively enhance their bandwidth efficiency in comparison with existing common public radio interface (CPRI)-based fronthauls [21]. It should also be noted that the proposed PONs are completely different from the multiband OFDM PON reported in [22], since the PONs proposed here have the following unique features: (1) each subwavelength spectral region is shared by two independent orthogonal OFDM sub-band signals [6]; (2) no extra channel spacing is required between adjacent subwavelengths or sub-bands; (3) the side lobes of each OFDM sub-band are considerably reduced by the digital filtering process, which can minimize the interchannel interference (ICI) effect between sub-bands at adjacent subwavelengths. Furthermore, our results also indicate that for subwavelengths that do not suffer the strong channel fading effect, the ICI effects between different orthogonal OFDM sub-bands in these subwavelengths are negligible; (4) as a direct result of using the digital filtering process, each OFDM sub-band can adaptively and flexibly adjust its signal modulation parameters—such as subcarrier count and channel bandwidth—without affecting the orthogonality between different sub-bands; (5) the digital-filtering-processing-induced ICI reductions greatly enhance the PON’s performance and its robustness against the channel frequency offset. In comparison with the up-conversion-based OFDM multiband PONs, which require multiple tuneable electrical local oscillators [20], the spectrally overlapped hybrid OFDMDFMA PONs utilize only digital filters to multiplex multiple OFDM sub-bands without requiring extra electrical/optical components compared to conventional transceivers in both the ONUs and the OLT.

## 2. Principle of DFT-Spread Spectrally Overlapped Hybrid OFDM–DFMA PONs

A representative DFT-spread spectrally overlapped hybrid OFDM–DFMA IMDD PON architecture is depicted in Figure 1, in which only the more challenging multipoint-to-point upstream operation is considered. The additional *K*-point DFT and IDFT block at each ONU and OLT, combined with a subcarrier mapper and de-mapper module, are shown highlighted in yellow. In each ONU, either in-phase (I) or quadrature-phase (Q) M-ary quadrature amplitude modulation-encoded data symbols are grouped into blocks, each containing *K* symbols. The *K*-point DFT operation is applied to spread the symbols into the frequency domain. Localized mapping is then utilized to map the symbols onto an *N*-point (*N* > 2(*K* + 1)) IFFT with *N*-subcarriers. As the first subcarrier is unused, the *K*-signal-carrying subcarriers occupy the first *N*/2 − 1 subcarriers; as such, after zero padding, the signal length is *K* ≤ *N*/2 − 1. To produce a real-valued DFT-spread OFDM signal, an *N*-point IFFT operation is applied after enforcing the Hermitian symmetry. Following the addition of the cyclic prefix (CP), each digitally encoded sub-band sample sequence is up-sampled by a factor of *M*×. After that, the digital shaping filters, after having fed the up-sampled sequence, generate a digitally filtered sub-band signal, which is passed through the DAC and then fed into an optical intensity modulator (IM) to perform electrical-to-optical (E–O) conversion. Similar to the treatment adopted in [7], the utilization of IM as the preferred light source in the numerical simulations is to completely eliminate the signal–signal beating interference (SSBI) effect [23]. It is noteworthy that, for practical implementation of the proposed PONs, different ONUs can use different wavelengths to transmit their OFDM sub-bands, provided that every two adjacent wavelengths have a minimum wavelength space of ~0.28 nm to effectively mitigate the SSBI effects [24].

After transmitting through an SSMF, in the OLT, the corresponding receiver DSP functions are as follows: signal detection by a photodetector (PD); signal digitization by an ADC; serial-to-parallel conversion; symbol timing alignment; CP removal; single *L*-point FFT operation, with *L* satisfying *L* = *M* × *N* for generating complex-valued frequency domain subcarriers utilizing the received real-valued time-domain symbols; identification of lower sideband (LSB) and upper sideband (USB) subcarriers; independent channel estimation and channel equalization of LSB and USB subcarriers based on the pilot subcarriers; and sideband processing to demultiplex the spectrally overlapped spread-spectrum OFDM subcarriers. After the above DSP process, the resulting output sub-band subcarriers’ data are passed through the subcarrier de-mapper. The output of the de-mapper is then subjected to the *K*-point IDFT and symbol demodulation to obtain data information corresponding to the transmitted I-phase and Q-phase ONUs’ sub-band input data.

## 3. Upstream Optimum ONU Operating Conditions

In performing the numerical simulations over a 25 km SSMF, an IMDD PON theoretical model developed and verified in [6] was adopted, where the procedure detailed in [25] was used to simulate the optical OFDM signal generation, nonlinear transmission, and direct detection.

### 3.1. Simulation Models and Key Parameters

In this paper, two ONUs are considered, each producing an optical sub-band signal sharing the same subwavelength with the other ONU. To conduct the simulations, MATLAB tools were used for signal generation and detection, while VPItransmissionMaker was used for optical fibre transmission. To implement the shaping filter in each individual ONU, a widely used digital filter construction approach called the Hilbert-pair approach was adopted to produce two orthogonal digital shaping filters sharing a subwavelength spectral region. These two orthogonal filters form a Hilbert pair, which have similar amplitude characteristics, but with a π/2 phase difference in the frequency domain [26,27,28]. To support two independent ONUs, the up-sampling factor was set at *M* = 2 [26]. Since the DAC/ADC operates at 12.5 GS/s and the two ONUs occupy the same spectral region, the signal bandwidth for each ONU is equal to the Nyquist frequency, *fs*/*M*, while the central frequency of the orthogonal digital filter pair is *fs*/2*M,* where *fs* is the DAC/ADC sampling speed. For the *M* = 2 case, Figure 2a,b illustrate the spectral locations of digitally filtered sub-band signals (I-phase and Q-phase) and their spectra. To generate the real-value OFDM signal necessary for intensity modulation, a 32-point IFFT size was considered, in which 15 subcarriers in the positive frequency bins convey real data, one subcarrier contains no power, and the remaining 16 subcarriers in the negative frequency bins are the complex conjugates of data-bearing subcarriers. To reduce the power leakage caused by crosstalk between spectrally overlapped digital orthogonal sub-bands, and maximize the upstream signal transmission capacity, 14 data-bearing subcarriers out of 15 were employed to deliver the acceptable upstream performance for each sub-band. Thus, the DFT block size *K* was set to be 14. It is, however, expected that all 15 of the subcarriers can be supported if channel interference mitigation techniques are applied [29,30]. In this demonstration, all of the subcarriers were encoded with a 64-QAM signal modulation format; however, any modulation formats are applicable.

Detailed explorations of the impact of quantization and clipping noise on the upstream transmission performance of the digitally filtered spread-spectrum OFDM signals are undertaken in Section 3.3, in which an optimal 7-bit resolution and optimal clipping ratios of 11 dB and 12 dB are identified for the cases of including and excluding the DFT-spread, respectively. These identified optimal parameters are adopted throughout this paper. Unless otherwise stated, all other system parameters are summarized in Table 1.

In the OLT, an ideal positive-intrinsic-negative (PIN) photodetector for direct detection of the optical signal is employed, with a receiver sensitivity of −19 dBm and a quantum efficiency of 0.8 A/W. Both shot noise and thermal noise are considered, and are simulated using procedures similar to those presented in [30]. In addition, the OLT-based receiver consists of a variable optical attenuator (VOA) to adjust the received optical power (ROP) level, while the ADC incorporates an ideal antialiasing electrical filter with a 6.25 GHz bandwidth to remove out-of-band receiver noise before signal sampling.

Taking into account the transceiver parameters listed in Table 1 and the adopted signal modulation formats, the upstream signal transmission rate per ONU is ~13.12 Gb/s, while the aggregate upstream PON transmission rate is ~26.25 Gb/s. It is noteworthy that, due to the high attenuation of the filter near the DC component, only 14 subcarriers (2–15) are activated in each ONU, as the first subcarrier contains no power.

### 3.2. PAPR Performance of DFT-Spread Hybrid OFDM–DFMA PON

Having chosen the simulation parameters, identified the optimal conditions, and understood the operating principle of the proposed DFT-spread spectrally overlapped hybrid OFDM–DFMA IMDD PON, in this section, we numerically explore the PAPR reduction efficiency of the proposed PON consisting of two ONUs. Figure 2a,b show the spectral locations of two digitally filtered sub-band signals (I-phase and Q-phase) and their spectra. In addition, the comparative complementary cumulative distribution functions (CCDFs) of the PAPR, including and excluding DFT-spread, are presented in Figure 2c,d, respectively, for various digital filter lengths ranging from 16 to 256 and signal modulation formats varying from 16-QAM to 256-QAM. For the sake of simplicity, only ONU-2—referred to as channel-2—CCDF curve is plotted for both cases, as the curves of ONU-1—referred to here as channel-1—are similar to those of channel-2.

It can be seen from Figure 2 that, compared to the case without DFT-spread, the case with DFT-spread can reduce the PAPR ratio by >2 dB at the CCDF value of 1 × 10^−3^ when the optimal clipping ratio of 11 dB is adopted. The results are similar to those of our previous work, and those observed in WDM-PONs [19,31,32]. It is also very interesting to note in Figure 2c,d that the proposed PON’s upstream transmissions have very similar PAPR performances when varying both digital filter lengths up to 256 and signal modulation formats up to 256-QAM. This indicates that the DFT-spread-induced PAPR reductions are independent of the digital filter length and the signal modulation. Moreover, the obtained results also suggest that, under the same transmission power constraints, the DFT-spread case with a low PAPR can achieve a higher optical signal-to-noise ratio (OSNR); this is one of the factors leading to the increased upstream channel rate presented in Section 4.2. Nevertheless, it is worth noting that the performance of the PONs reported in [6] is largely limited by digital-filter-induced signal distortion. As such, in this paper, to highlight the unique features of the DFT-spread technique, a digital filter length of 64—as listed in Table 1—is utilized to minimize the digital filter impairments.

From the above discussion, it is easy to understand that the proposed PON-induced PAPR reduction gives rise to an excellent improvement in system performance robustness to quantization noise induced by limited DAC/ADC bit resolutions. In addition, it also relaxes the constraints on the linear dynamic operating ranges of the transceiver-embedded optical/electrical devices, reduces the optical nonlinearity impairments, and allows reductions in the DSP complexity and overall cost of the transceivers.

### 3.3. Optimum Clipping Ratio and DAC/ADC Resolution Bits

In order to numerically explore the feasibility of utilizing the proposed technique to improve the upstream transmission performance of the PON, in this section, numerical simulations are first undertaken to identify the optimal operating conditions for achieving the best possible performance. Figure 3 presents the simulated bit error rate (BER) contours as a function of quantization bit and clipping ratio (CR) for an optical back-to-back (B2B) configuration. In obtaining these figures, the ROP at the OLT is fixed at −7 dBm. Since the two ONUs are independent and have the same signal characteristics, without loss of generality, only channel-2 is plotted in Figure 3.

It can be seen in Figure 3 that when the CR is low, the overall channel BER performance is high because the signal waveform is significantly clipped. In these figures, for a fixed bit resolution within a dynamic range from 7 to 8 bits, to maintain BERs below the FEC limit (1 × 10^−3^), the CR should be ≥11 dB for the DFT-spread case. On the other hand, for the case without DFT-spread over the same dynamic region, the CR should be ≥12 dB, which shows an increase in CR by 1 dB. From the same figures, it can also be observed that the case excluding DFT-spread has a CR dynamic range of 12 dB ≤ CR ≤ 14 dB, while the case including DFT-spread achieves a CR dynamic range of 11 dB ≤ CR ≤ 14 dB. Outside these regions, the BER increases with increasing CR because of the rise in the quantization noise effect, and decreasing the CR causes the creation of in-band/out-band noise due to clipping distortion. This result indicates that the proposed PON can reduce the DAC/ADC bit resolution by 1 bit—i.e., from 7 bits to 6 bits—and reduce the CR by ≥1 dB to achieve BERs within the FEC limit. This behaviour clearly demonstrates that the proposed PON with the application of the DFT-spread technique allows the transceiver to adopt low CRs, without greatly compromising the BER performance. Based on the above analysis, the identified optimal CR values and a DAC/ADC resolution of 7 bits were chosen to enable the ONUs to operate at their optimal conditions. The obtained results for the optimal parameters are presented in Table 1.

## 4. Upstream DFT-Spread Hybrid OFDM–DFMA PON Performance

Utilizing the optimal ONU operating conditions identified in Section 3 and the transceiver parameters listed in Table 1, in this section, the investigations of the upstream transmission performance of a spectrally overlapped hybrid OFDM–DFMA PON incorporating the DFT-spread technique are undertaken in terms of upstream performance tolerance to limited DAC/ADC quantization bits, BER performance, maximum aggregate upstream signal transmission rate, and impact of digital filter impairments. To highlight the advantages associated with the proposed technique, the upstream transmission performances of the hybrid OFDM–DFMA PON utilizing spectrally overlapped digital orthogonal filtering are also computed, and are treated as benchmarks.

### 4.1. Performance Tolerance to Limited DAC/ADC Quantization Bits

In this subsection, simulations are carried out to demonstrate the performance tolerance of the proposed technique to limited DAC/ADC quantization bits, and to determine the minimum number of required DAC/ADC quantization bits to achieve BERs at or below the FEC limit. The DAC/ADC quantization bits vary from 4 to 8, and the ROP is fixed at −7 dBm. The results are presented in Figure 4 for the 25 km SSMF IMDD PON.

Figure 4 reveals that, while adopting the optimal CRs of 11 dB or 12 dB as determined in Section 3.3, the case including DFT-spread can reach BERs at the FEC limit when the DAC/ADC resolution is as low as 6 bits. On the other hand, for the case excluding DFT-spread, the minimum number of required quantization bits to achieve similar performance extends from 6-bit to 7-bit DAC/ADC resolution. This numerical result confirms that the application of DFT-spread in the proposed PON can improve the upstream performance tolerance against quantization noise induced by the limited quantization bits. Most importantly, from a practical PON operation point of view, such an improvement is highly desirable for PON designs, as lower DAC/ADC hardware achieves both lower cost and lower power consumption.

From the above discussion, it is easy to understand that the proposed DFT-spread spectrally overlapped hybrid OFDM–DFAM PON confirms the practicability of utilizing the DFT-spread technique in the PONs reported in [6], and that the proposed PON has the ability to reduce the minimum required DAC/ADC bit resolution and, consequently, minimize the transceiver DSP complexity, thus making the proposed PON a promising solution for implementation in future cost-sensitive 5G networks, and beyond.

### 4.2. Upstream Transmission Performance

Figure 5 presents the overall channel BER as a function of the received optical power, where the total optical launch power into the SSMF transmission system is fixed at 0 dBm. In computing Figure 5, a resolution of 7 bits was used for the ADC/DAC, as shown in Figure 4. All other parameters are specified in Table 1.

It can be seen in Figure 5 that, for the case of including/excluding DFT-spread, the calculated channel-1 (CH1) and channel-2 (CH2) BER performances are identical across the entire received optical power range, and these two channels also have very similar signal bit rates. Most importantly, for the same signal bit rates, the case with DFT-spread can achieve an upstream power budget improvement of more than 1.4 dB compared to the case without DFT-spread. Such performance improvement is mainly because of the DFT-spread-induced PAPR reduction, resulting in a considerable decrease in quantization noise, thus giving rise to an increase in the effective OSNRs.

### 4.3. Impacts of Digital Filter Parameters on Maximum Aggregated Upstream Transmission Rates

For the proposed PON simultaneously supporting two ONUs, the aggregated upstream signal transmission capacities versus the excess bandwidth parameter, α, are shown in Figure 6. In obtaining Figure 6, the digital filter length listed in Table 1 was adopted. In addition, adaptive bit-loading was applied to all subcarriers involved in each ONU for all of the cases of including and excluding DFT-spread. To implement adaptive bit-loading, the highest possible signal modulation formats within the range from DBPSK to 256-QAM were adaptively selected according to the channels’ spectral characteristics to ensure that the BER across all subcarriers for each sub-band could reach the FEC limit of 1 × 10^−3^. For each ONU sub-band, the achievable signal bit rate was $R_{b}=f_{s}\overset{N_{s}}{\underset{k=1}{\sum }}n_{kb}/2\left(N_{s}+1\right)\left(1+C_{p}\right)M$, where $n_{kb}$ is the number of binary bits conveyed by the *k*th subcarrier within one OFDM symbol period, $N_{s}$ denotes the number of data-bearing subcarriers, and $C_{p}$ indicates the overhead parameter associated with the cyclic prefix and training sequences. The excess bandwidth, α, was set to vary in the range 0 ≤ α ≤ 1. The ROPs were fixed at −6 dBm for both cases.

It can be seen in Figure 6 that, for both considered cases, the aggregate upstream signal transmission rate peaks at an α factor value of 0.2, and then reduces steadily as α increases above 0.2. Most importantly, for the case including DFT-spread, the maximum aggregate upstream signal transmission rate increases by up to 10% for α = 0.2, compared to the case without DFT-spread. This performance enhancement is due to the overall improvement in SNRs across the subcarriers, thus allowing higher modulation formats to be used. To understand the physical mechanisms causing an optimum α value of 0.2, the impact of α on the digital filter responses should be observed. For low α values < 0.2, the finite filter-length-induced filter magnitude response ripples can impact on performance. For α = 0 and the up-sampling factor *M* = 2, the I-phase digital filter has a perfectly flat response [26], which is equivalent to a case where the digital filter length (channel-1) is reduced to one, while the Q-phase filter (channel-2) has significant frequency response ripples, as shown in Figure 7a,b, respectively [26]. The length of the filter mainly impacts the sharpness of the filter edge in the case of the Q-phase filter. On the other hand, as shown in Figure 7c,d, as α starts to increase, there is a boost in the magnitude response of the I-phase filter and an attenuation in the magnitude response of the Q-phase filter. In addition the increasing distortion of the filter responses causes strong unwanted cross-channel-induced interference due to loss of orthogonality. It should be noted that the filter response distortions occur due to the aliasing effect as the employed subwavelength occupies the whole Nyquist band, and a narrower digital filter bandwidth can be observed when α > 0 [26]. The increasing α value also reduces the unwanted frequency response ripples in the Q-phase filter. These effects combine to give an overall increase in subcarrier SNR as α increases from 0 to 0.2, beyond which there is an overall reduction in subcarrier SNR.

To observe the effect of α on subcarrier SNRs, with the simultaneous presence of both channels, the SNR performances for all subcarriers for the 25 km SSMF transmission are plotted in Figure 7e, utilizing the optimum α parameter value of 0.2 and zero excess bandwidth (α = 0) for performance comparisons. In obtaining this figure, the ROP was fixed at −6 dBm. Figure 7e shows that the SNRs are similar for CH1 when α = 0 and α = 0.2; however, there is an obvious improved SNR for subcarrier indices 12–14 of CH2 when α = 0.2. Figure 7 also shows that the SNR curve for CH1 is almost constant with respect to the subcarrier index for the optimal α value. In comparison with the Q-phase channel for both cases, the high SNR observed for the I-phase channel is mainly a result of the boost in the magnitude response, as shown in Figure 7c. The obtained results indicate that there is an optimal α factor value that minimizes digital filter impairments and, subsequently, maximizes the upstream transmission rate of both channels.

## 5. Conclusions

The novel DFT-spread spectrally overlapped hybrid OFDM–DFMA PON was proposed and numerically simulated for a 25 km IMDD SSMF transmission system. To the best of our knowledge, this is the first work applying the DFT-spread technique to a hybrid OFDM–DFMA PON utilizing spectrally overlapped digital orthogonal filtering, in order to simultaneously reduce PAPRs, optimize CRs, and reduce the minimum required DAC/ADC quantization bits, whilst maintaining the upstream transmission performances and increasing the transceiver complexity. The simulation results showed that, compared to the previously reported PONs, the proposed PON can reduce the PAPR by more than 2 dB, and the optimal CR is reduced by 1 dB. Such a PAPR reduction is independent of the adopted digital filter characteristic, signal modulation format, and ONU sub-band signal spectral location. As a direct result of the PAPR reduction, the proposed PON can reduce the minimum required DAC/ADC resolution bits by more than 1 bit, whilst achieving BERs below the FEC limit. In addition, the proposed PON can improve the upstream power budget by more than 1.4 dB and increase the aggregate upstream signal transmission rate by up to 10%.

It is well known that the speeds of DACs and ADCs play important roles in limiting the maximum achievable signal transmission capacity for DSP-based signal transmission techniques. However, high-speed DACs and ADCs can be very expensive. To achieve a specific signal transmission capacity, the proposed technique has the potential of allowing low-speed DACs/ADCs to be utilized to produce/receive targeted baseband signals, which can then be up-converted to the targeted sub-bands by low-cost electrical components. This could significantly reduce the overall ONU transceiver cost. In addition, the proposed-technique-induced PAPR reductions also considerably relax the requirements of using expensive electrical/optical devices with large linear operation regions and high-resolution DACs/ADCs. As a direct result, the proposed technique may offer a valuable solution capable of reducing the overall network installation cost for implementation in cost-sensitive application scenarios.

The experimental demonstrations of point-to-point and multipoint-to-point spectrally overlapped hybrid OFDM–DFMA PON transmissions with/without DFT-spreading are currently being undertaken in our research laboratory, and the corresponding results will be reported elsewhere in due course.

## Figures and Tables

**Figure 1 sensors-21-05903-f001:**
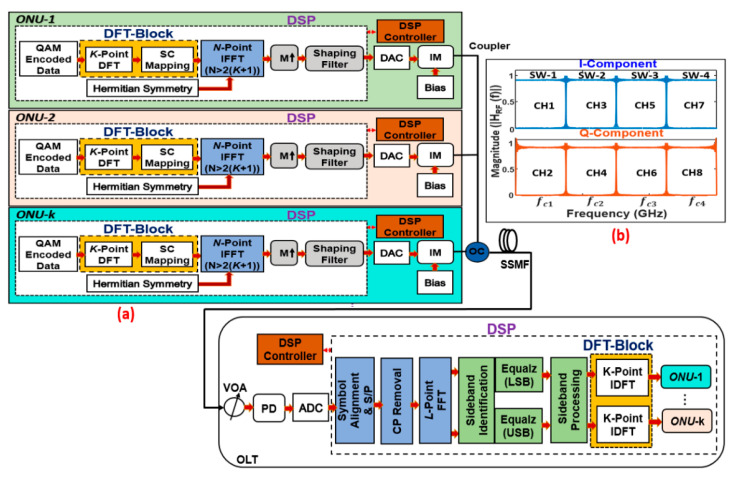
(**a**) DFT-spread hybrid OFDM–DFMA PON. (**b**) A schematic example of the ONU frequency planning and digital filter frequency responses for an excess of bandwidth of α = 0. SC: subcarrier; DFT: discrete Fourier transform; M↑: up-sampling factor; DAC/ADC: digital-to-analogue/analogue-to-digital converter; IM: intensity modulator; OC: optical coupler; SSMF: standard single-mode fibre; VOA: variable optical attenuator; PD: photodetector; S/P: serial-to-parallel conversion; Equalz: equalization; ONU: optical network unit; OLT: optical line terminal; DSP: digital signal processing; SW: subwavelength; CH: channel.

**Figure 2 sensors-21-05903-f002:**
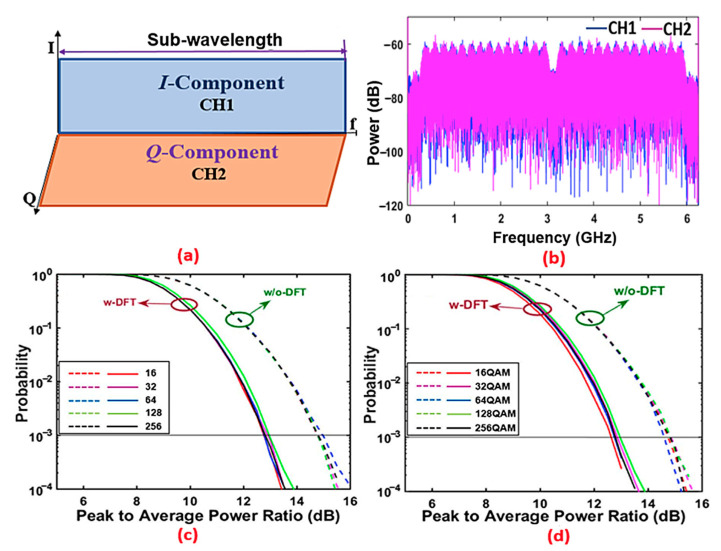
(**a**) Spectral location of orthogonally digitally filtered sub-band signals (I-phase and Q-phase) and (**b**) their spectra. CCDFs of PAPR for (**c**) varying digital filter lengths and a fixed 64-QAM modulation format, and (**d**) varying modulation formats and a fixed digital filter length of 64.

**Figure 3 sensors-21-05903-f003:**
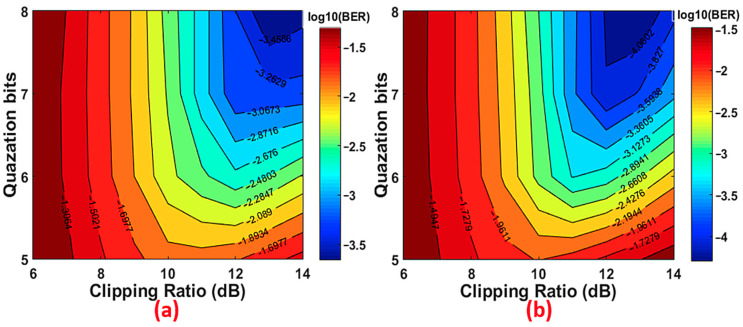
BER contour versus quantization bit resolution and clipping ratio for (**a**) excluding the DFT-spread, and (**b**) including the DFT-spread.

**Figure 4 sensors-21-05903-f004:**
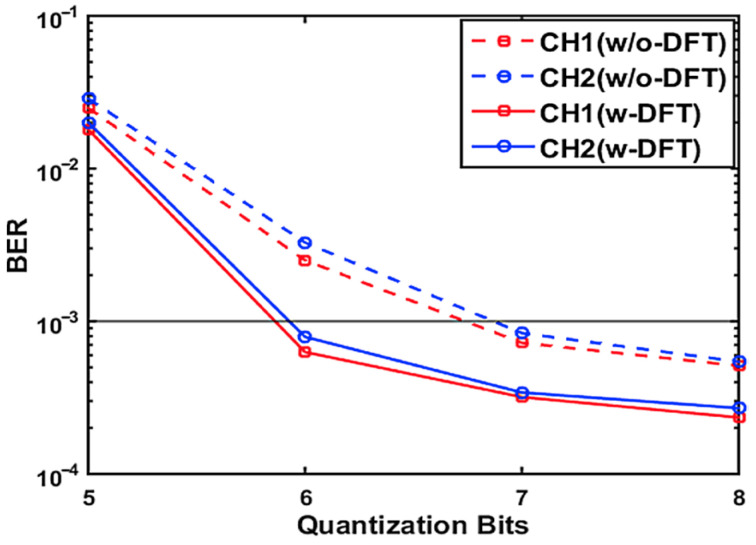
BER versus quantization bits over a 25 km SMMF IMDD PON.

**Figure 5 sensors-21-05903-f005:**
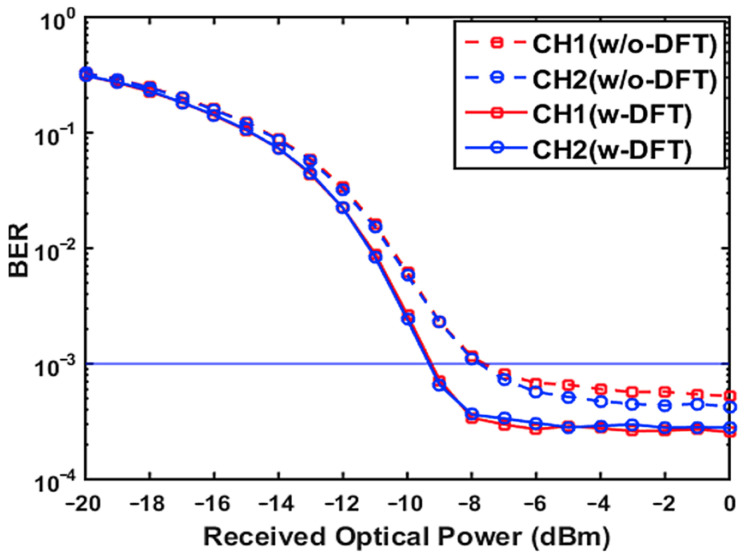
BER versus received optical power over a 25 km SSMF IMDD PON transmission system when a 7-bit resolution is considered.

**Figure 6 sensors-21-05903-f006:**
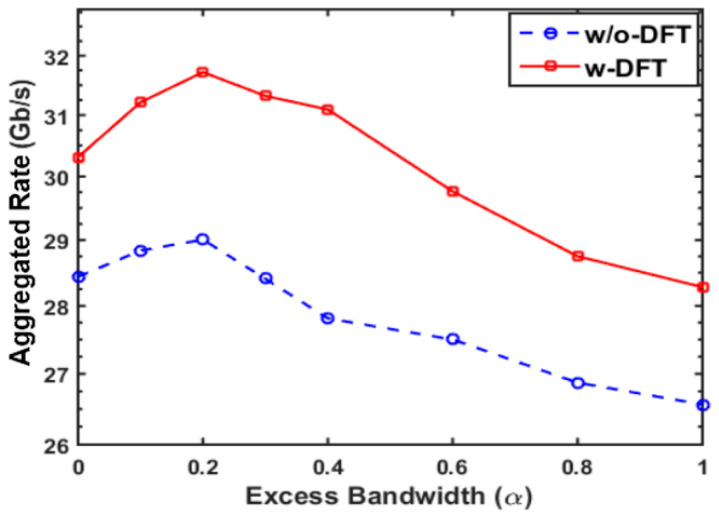
Aggregated rate versus excess bandwidth over a 25 km SSMF IMDD PON transmission system when a 7-bit resolution is considered.

**Figure 7 sensors-21-05903-f007:**
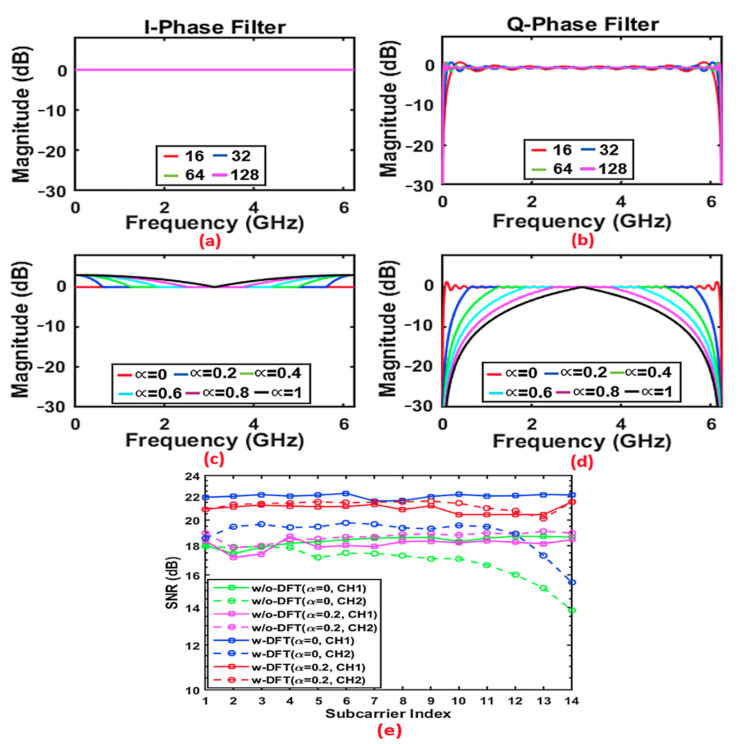
Magnitude response for excess of bandwidth of α = 0 for (**a**) I-phase filter with digital filter lengths 16, 32, 64, and 128; (**b**) Q-phase filter with digital filter lengths 16, 32, 64, and 128. Magnitude response for filter length 64 for (**c**) I-phase filter with excess bandwidth (α) ranging from 0 to 1, and (**d**) Q-phase filter with excess bandwidth (α) ranging from 0 to 1. (**e**) Comparison of SNR distribution across all of the subcarriers, using excess bandwidth of α = 0 and an optimal value of α = 0.2 for both I-phase (CH1) and Q-phase (CH2).

**Table 1 sensors-21-05903-t001:** System parameters.

Parameter	Value	Parameter	Value
IFFT/FFT Size	32/64	Clipping Ratio—Including /Excluding DFT-spread	11 dB/12 dB
Number of Used Data Subcarriers per ONU	14	Digital Filter Length/Excess Bandwidth	64/0
Modulation Format	64-QAM	PIN Detector Quantum Efficiency	0.8 A/W
Cyclic Prefix	25%	PIN Detector Sensitivity	−19 dBm
Channel Bitrate	13.12 Gb/s	PIN Detector Bandwidth	Ideal
Optical Launch Power	0 dBm	Fibre-Dispersion	17 ps/nm/km
DAC/ADC Sample Rate	12.5 GS/s	Fibre-Dispersion Slope	0.08 ps/nm^2^/km
Number of Bits	7-bits	Fibre Loss	0.2 dB/km
Up-sampling Factor	M = 2	Fibre Kerr Coefficient	2.6 × 10^−20^ m^2^/W
FEC Limit ^1^	1 × 10^−3^	Transmission Distance	25 km

^1^ Corresponding to 10 Gb/s non-return-to-zero data at a BER of 1.0 × 10^−9^.

## Data Availability

Not applicable.

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
