# Peer review of "DFT-Spread Spectrally Overlapped Hybrid OFDM–Digital Filter Multiple Access IMDD PONs"

_sensors, 2021, doi:10.3390/s21175903_

Round 1
Reviewer 1 Report
As I understand, the paper presents the results of investigations following those presented in Ref. 6. The novelty of the current paper with respect to Ref. 6 is the application of DFT spreading (or precoding) to the transmitted data using the OFDM method. The authors present the consequences (positive changes) when DFT precoding is applied as compared with the case when it is not applied (as in Ref. 6). In general, the paper is interesting and well written. However, some improvements are recommended.
- I propose the English co-authors to read the paper carefully, as I have found several small grammar errors in it. They can be easily elilminated by native speakers.
- The authors report the results related to the applied digital filters in the receiver. One cannot find any functional block denoting these filters in the block diagram. It could be much better when such a block can be seen in Fig. 1.
- The idea of DFT precoding in OFDM transmission is not new. It has been applied in uplink of LTE system and is an option in uplink of 5G New Radio. In this context I am slightly disappointed why the difference in CCDF between the case when DFT precoding is applied and when it is not is about 2 dB only.
In generaal, after proposed changes and added comments the paper will be publishable.
Reviewer 2 Report
This paper presents an incremental improvement following a long sequences of papers by some of the authors. There are certain technical merits that are worthy of readership. However, there are several serious issues need to be addressed.
- I am very confused by the naming practice. The proposed technique is a supposed "DFT-Spread Spectrally Overlapped Hybrid Orthogonal OFDM-Digital Filter Multiple Access IMDD PON." I mean, what is this? I have to go through many of the authors' past publication to get a slim idea of, say, what does "hybrid", "spectrally overlapped", and "orthogonal OFDM-digital filter" mean. And frankly, I don't agree with the choice of those terms. For instance, the term "spectrally overlapped" seems to describe the co-existence of I and Q channels. (As a matter of fact, I am not 100% sure, due to the vagueness of how the authors presented their results in the past.) That's quadrature multiplexing, and it is so common in passband modulation that it should not appear at all.
- The paper is mostly an incremental improvement based on [6]. However, the description on the common part and the differences is very insufficient. That unnecessarily increases the burden of readers. I have to come back and forth between two papers frequently to understand what exactly is presented.
- DFT-spreading is a well-known technique, as it also appears in the 4G LTE standard. It is worthwhile to see how the technique fares in fiber optics. But the presentation needs to be much clearer for readers to get the concept. For instance, OFDM introduces the idea of subcarriers and DFT (BTW, I prefer using DFT instead of FFT; FFT is just an implementation technique) maps signals on subcarriers, while the authors also use "sub-wavelength" and "subband", it is very important to present these terms in a clear way so that no confusion is made.
- In 4.2, an I-phase "perfectly flat" filter is mentioned. How could this be possible? More explanations are desperately needed. The only possibility is that the filter actually does nothing. But then, how can the filter limit the signal's bandwidth?
- With such high bandwidth, all-digital solutions can be very expensive. I wish the authors can comment on that.
- I have also attached a file to highlight the places where I found confusing.

Round 2
Reviewer 2 Report
Most of my concerns have been addressed. I am ok if the authors insist using a very lengthy and non-informative name for their technique. Indeed it's not friendly to readers.
Three more places need fixing before publication.
- In Fig. 1(b), the "frequency responses" of digital filters are shown. My understanding is that it is only applicable for the case where the roll-off factor is 0. It would be more meaningful to show explicitly possible frequency planning of ONU's or even OFDM subcarriers, instead of this vague indication of what has been done. (BTW, I don't understand why the authors use the Hilbert pair to refer to the pair of quadrature filters. It's simply not a common name and unnecessarily increases readers' burden.)
- In Fig. 7(a) a flat response is shown. The explanation is that "the I-phase filter (channel 1) has a perfectly flat response as the up-sampling factor M=2 is considered [26]". Well, it would be better if the authors inform readers that in this situation, the I-phase filter reduces to a unit-tap one (which means doing nothing).
- In Fig. 7(d) The filter with a larger roll-off factor has a smaller bandwidth. So, the definition of excess bandwidth is different from what is usually seen? Why?
Author Response
The authors deeply appreciate the reviewer's contributions for improving the quality of this MS. Please see the attachment for the response.
